# Flexible-Imaging-Fiber-Guided Intratracheal Intubation in Rodents

Sang Hoon Jeong [1,2,†], Cherry Kim [3,†], Hong Lee [2], Yoon Jeong Nam [2], Chil hwan Oh [1,4], Yong-Wook Baek [5], Jungyun Lim [5], Ju-Han Lee [6,*] and Jaeyoung Kim [1,2,7,8,*]

1 Research Institute for Skin Image, Korea University College of Medicine, Seoul 08308, Republic of Korea
2 Medical Science Research Center, Ansan Hospital, Korea University College of Medicine, Ansan-si 15355, Republic of Korea
3 Department of Radiology, Ansan Hospital, Korea University College of Medicine, Ansan-si 15355, Republic of Korea
4 Department of Dermatology, Wonkwang University School of Medicine, Iksan-si 54538, Republic of Korea
5 Humidifier Disinfectant Health Center, Environmental Health Research Department, National Institute of Environmental Research, Incheon 22689, Republic of Korea
6 Department of Pathology, Ansan Hospital, Korea University College of Medicine, Ansan-si 15355, Republic of Korea
7 Department of Dermatology and Skin Science, University of British Columbia, Vancouver, BC V6T 1Z1, Canada
8 Departments of Cancer Control Research and Integrative Oncology, British Columbia Cancer Agency, Vancouver, BC V5Z 1L3, Canada
* Correspondence: repath@korea.ac.kr (J.-H.L.); jaykim830@gmail.com (J.K.); Tel.: +82-31-412-5322 (J.H.L.); +82-31-412-6711 (J.K.)
† These authors contributed equally to this work.

**Abstract:** Although experiments on intratracheal intubation for animals are essential for research, it remains challenging. This study aimed to validate an animal model using a flexible imaging guide system that can be conveniently and safely used as a new method to provide easy access to organs in small animals. PBS (Phosphate Buffered Saline) and PHMG (Polyhexamethylene guanidine) were administered by intratracheal intubation to 20 rodents (10 mice and 10 rats), and the changes in the lungs were observed. Results were verified using lung tissue histopathologic staining through the intratracheally administered material, which confirmed that 100% of changes in lung tissue occurred in the PHMG-injected group, where intubation was facilitated using the flexible imaging guide. The drug was conveniently and safely administered. The flexible-imaging-fiber-guide-based intratracheal drug injectable system may be conveniently used by researchers. It allows drugs to be administered quantitatively, suggesting its potential wide use in drug development and toxicity evaluation.

**Keywords:** flexible imaging guide; intubation; PHMG

## 1. Introduction

The severity of health issues caused by air pollution and chemicals has steadily increased with economic growth and industrial development. As stated in the WHO report, air pollution kills approximately seven million people worldwide each year. According to WHO data, nine out of ten people in low- and middle-income countries are exposed to high levels of pollutants [1].

In South Korea, Polyhexamethylene guanidine (PHMG) came under scrutiny after several deaths and illnesses were linked to humidifier disinfectants containing PHMG in 2011. The outbreak resulted in the death of over 100 people, with hundreds more suffering from lung injuries and other respiratory problems [2]. The incident prompted the Korean government to investigate the safety of humidifier disinfectants and other products containing PHMG. The research conducted in South Korea on the impact of PHMG on human health has been extensive. The Korean government set up a task force to investigate the issue, which included experts from various fields, such as toxicology, pulmonology, and epidemiology. The task force conducted multiple studies to understand the health

effects of exposure to PHMG-containing products and identify the factors contributing to the outbreak. One of the most significant studies was a case-control study investigating the association between exposure to humidifier disinfectants containing PHMG and lung injury. The previous study found a strong association between disinfectant exposure and lung injury, with risk increasing with increased exposure. The study identified that the disinfectants caused severe lung inflammation, leading to respiratory failure [3].

Other studies conducted by the task force investigated the toxicity of PHMG in animal models and the mechanism of PHMG-induced lung injury. These studies identified that PHMG exposure could cause significant damage to the lungs, liver, and kidneys [2]. The studies also recognized that the mechanism of PHMG-induced lung injury involved the activation of immune cells in the lungs, leading to severe inflammation and tissue damage [4]. The research conducted in Korea on PHMG has shown significant changes in regulations and guidelines for disinfectants and other products containing PHMG. The Korean government banned producing and selling humidifier disinfectants containing PHMG and other harmful chemicals [5]. The government also introduced stricter regulations on the use of disinfectants in public places such as schools and hospitals.

Air pollution has several adverse health effects. Short-term exposure to air pollutants is closely related to respiratory diseases such as chronic obstructive pulmonary disease and cough [6]. However, to elucidate the causal relationship between harmful substances and diseases, there is a need for long-term research and support from experimental subjects [7,8]. However, it is not possible to conduct research on respiratory diseases or their etiology by testing toxic substances directly on humans. Such studies should, therefore, be conducted in preclinical studies.

The most challenging aspect of animal models is the precise and reproducible evaluation of lung disease or treatment by injecting multiple substances into the lungs. Research on disease diagnosis and treatment through establishing models of chronic obstructive pulmonary disease (COPD) is steadily increasing. COPD is the primary cause of morbidity and mortality worldwide and is characterized by chronic airway inflammation, mucus hypersecretion, airway remodeling, and emphysema, resulting in decreased lung function and dyspnea [9]. Gradual worsening of lung function due to aging and exposure to various harmful materials may progress to COPD [10].

Rodent models are widely used in research to investigate and establish a disease model for studying lung diseases, such as COPD. These animal models are widely used because they are economical and have good reproductive power [11]. These include inhalation and intratracheal intubation with drugs. Drug injection through inhalation is widely used in research on inhaled substances, as no animal sedation or anesthesia is required [12]. However, this requires expensive equipment and has the disadvantage of low mass-transfer efficiency [13,14]. It is difficult to accurately control and determine the amount delivered to the lungs because the inhaled material is deposited in the nasopharynx during respiration, which is different from the human oral structure. Moreover, rodent models inside the chamber are under significant stress during administration due to limited food and water supply. Intratracheal intubation usually involves direct introduction into the airway.

Tracheostomy is a surgical procedure in which the trachea is cut, and the substance is administered directly to the lungs through the airway. This is commonly used for pulmonary delivery in rodent models because of its minimal drug loss and high delivery efficiency upon intratracheal administration [15]. Tracheostomy is a surgical procedure that creates an opening in the trachea to facilitate breathing or to allow direct access to the respiratory system for drug or toxin administration. The use of tracheostomy for drug or toxin administration has advantages and disadvantages that must be carefully evaluated. The benefits of tracheostomy are as follows. Tracheostomy allows direct access to the respiratory system, which is particularly useful for drug or toxin administration targeting the lungs. This method ensures that the substance of interest reaches the lungs efficiently and rapidly, bypassing other organs and tissues that may otherwise metabolize or eliminate it before reaching the target site. It allows for precise dosing of the administered substance,

as it can be directly delivered to the respiratory system without any loss or degradation. It can reduce the risk of systemic toxicity associated with some drugs or toxins by avoiding exposure to other organs and tissues, thus minimizing unwanted side effects [16].

However, tracheostomy is a surgical procedure that carries some risk, particularly in rodents, and its disadvantages include high side effects for microsurgery and repetitive surgeries. There is a risk of bleeding, infection, or other complications associated with the procedure, which could impact the animals' well-being and affect the study results. It can cause respiratory distress in rodents, particularly in the early postoperative period. This distress can affect the animals' behavior, physiology, and overall health, potentially confounding the study results.

This paper describes an imaging-based intubation system through which the desired animal model can easily be implemented by visually checking the vocal cord and administering several drugs or toxins through the route. This study, therefore, provides information on the procedure and system for injecting drugs directly into the lungs of rodent models.

## 2. Materials and Methods

### 2.1. Animals

A total of 20 rodent models were used in this study. Eleven Sprague–Dawley strain nine-week-old male rats (250–300 g) and eleven C57BL/6 strain six-week-old male mice were purchased from Raonbio Korea (Yong-in, South Korea). The rodent models were acclimated for one week. The following conditions were set: temperature of 22–25 °C; relative humidity of 40–60%; and lighting conditions of 12 h light alternating with 12 h dark. Food pellets (Purina, Sung-nam, South Korea) and filtered tap water were provided for the experimental rodents ad libitum. Autoclaved bedding was provided to each cage and changed once per week.

This study was approved by the Institutional Animal Care and Use Committee of Korea University Medical Center (approval number: Korea-2019-0031-C2, Korea-2021-0124-C1).

### 2.2. Flexible-Imaging-Fiber-Guided Intratracheal Administration System

Figure 1 presents a schematic of the proposed flexible fiber imaging-based toxic substance dosing system. The proposed system comprised an inhalation system for continuous substance administration under respiratory anesthesia, a scientific camera for acquiring high-quality intratracheal images, and a peristaltic pump for transporting liquid and aerosol substances. The imaging system visualized the intratracheal site of the rodents based on soft fibers. When the tip of the injection system was located inside the upper airway, the material was gently administered by driving the pump at a constant speed. The intubated tube was removed five seconds after administration.

Equipment List

The imaging system is composed of a fiber bundle for imaging; a CMOS camera (Thorlabs DCC3240C; 1024 × 1280 pixels, pixel size: 5.3 μm); an optical fiber bundle connector (Orient Medical; C-mount Camera Coupler Zoom F18-F35, Hangzhou, China); a light source (Kwang Fiber Tech; KFT-L75, South Korea)) connected with a fiber bundle; a drug intubation tube and a pump (DFROBOT; Digital Peristaltic Pump, DFR0523, South Korea); and a flexible imaging part composed of a 10,000-pixel fiber bundle and 0.75 mm fiberscope diameter with illumination (FIGH-10-350S; Myriad Fiber Imaging, Dudley, MA, USA). The total outer diameter of the imaging fiber was 1 mm, and the outer diameter of the core fiber bundle was 0.75 mm.

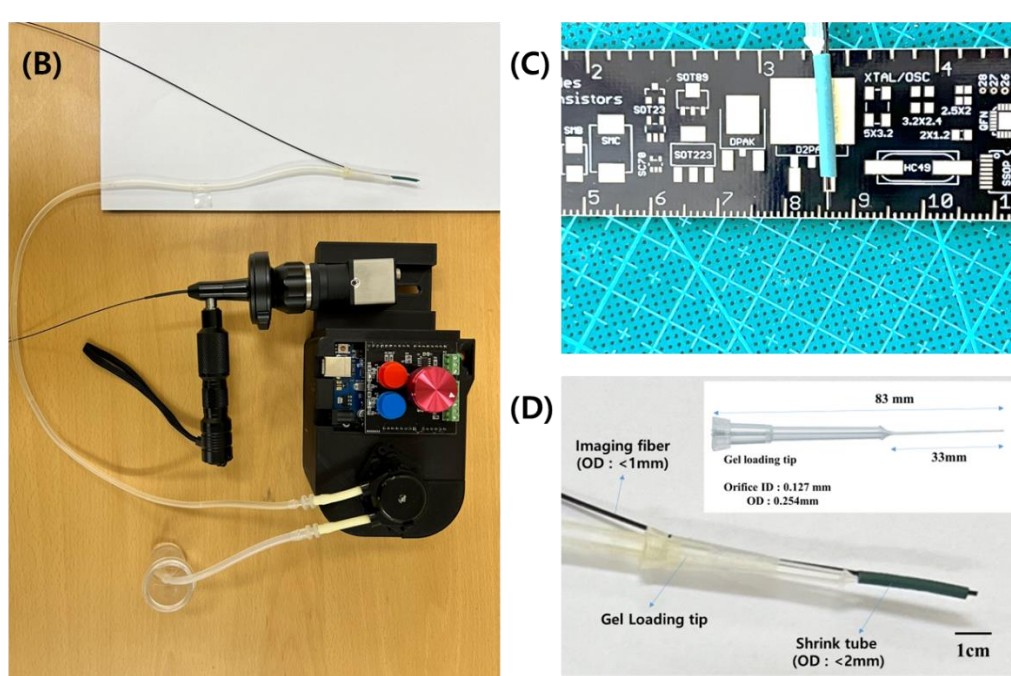

**Figure 1.** The proposed Flexible-Imaging-Fiber-guided intratracheal administration system. (**A**) schematic diagram of the flexible intratracheal imaging and administration system, (**B**) an image of the system, and (**C,D**) enlarged image of the video and drug administration probe. (OD: outer diameter, ID: inner diameter, <: less than).

*2.3. Pulmonary Administration of the Polyhexamethylene Guanidine (PHMG)*

Anesthesia was initially induced by placing the rodent models in a chamber filled with 4–5% volume-to-volume for four to six minutes. The models were anesthetized with isoflurane and placed on a holding stage (Figure 2) with the inhaled isoflurane concentration fixed at 3%.

Ten SD strain rats and ten C57BL/6 mice were randomly divided into four groups (five rodents in each group). An aqueous solution of PHMG (CAS No. 89697-78-9) was prepared by diluting it to 0.9 mg/kg with saline buffer (PBS, Thermo Fisher Scientific Inc., Waltham, MA, USA) as per previous studies [17–19]. During respiration, the vocal cords in rodents (and all mammals) play an essential role in regulating airflow into and out of the lungs. When the rodent inhales, the vocal cords relax and move apart, creating a wider opening for airflow (Figure 3). We administered the test substance through Gel loading tips or an intubation tube inserted into the intratracheal when they inhaled air. In more detail, 50 μL of PHMG aqueous solution was administered intratracheally by inserting a microtube for injecting the drug when the vocal cords were completely opened while simultaneously monitoring the larynx region through a flexible fiber imaging device. For

the control group, 50 µL of saline buffer, the same PHMG mixture, was intratracheally administrated to each group with the proposed imaging guide system.

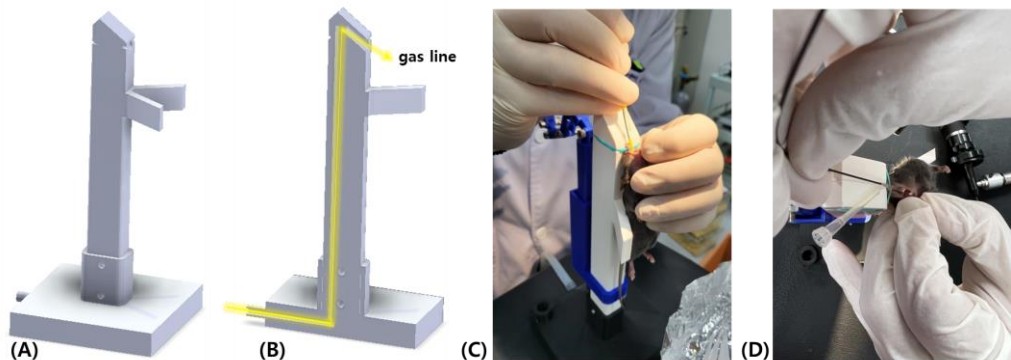

**Figure 2.** The three-dimensional printed inhalation anesthesia system and fixation stage for rodent models. (**A**) The three-dimensional design of a respiratory anesthesia and rodent holder to assist a drug or toxic substance administration system. (**B**) Half-section view of the system and holder for rodents from the three-dimensional design. (**C**) perspective view and (**D**) plan view pictures of drug or toxic substance administration experiment with a flexible imaging guide after rodent fixation.

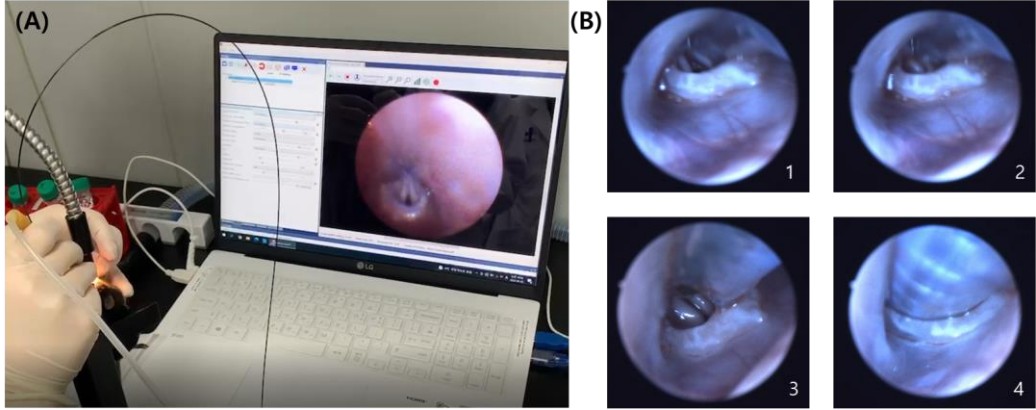

**Figure 3.** Intubation result image based on the manufactured system. (**A**) An experiment using an intubation system. (**B**) Intubation results in the images in rodent models. (B-1: opened vocal cord, B-2: closed vocal cord, B-3: little opened vocal cord, B-4: the inserted intubation tube when the vocal cord was opened).

*2.4. Histologic Examination*

The changes in the lungs of all animal models were monitored, and the lung tissue was isolated and fixed in 10% neutral-buffered formalin after sacrifice. Subsequently, four µm-thick paraffin sections were cut from the fixed tissues, and hematoxylin and eosin (H&E) staining was performed. Five to six tissue sections were examined per model (at least one or two sections per lung lobe of each rat). The presence and number of bronchoalveolar hyperplasia and abnormal changes were evaluated in each group.

**3. Results**

The animals were anesthesia induced in the closed chamber. Once adequate anesthesia was achieved, the animals were placed on an inhalational anesthesia stand. To open the mouth, a rubber band was placed on the front teeth of the upper jaw to secure it, and the tongue was carefully pulled away by the observer using forceps. The proposed imaging-based delivery system was positioned inside the mouth to clearly visualize the tracheal site. The cannula tip of the administration system was then inserted into the vocal cord position while keeping an eye on the video. A drug tube filled with PHMG substance in

liquid form was injected with an appropriate amount of air using a pump. In this study, the PHMG injection model of ten animals (mice = five, rats = five) was compared with the PBS injection model of ten animals (mice = five, rats = rive). As a result, the inflammatory cells were monitored in PHMG injection groups. Proliferation was confirmed, and no abnormal lesions were detected in PBS-injected groups.

PHMG solution was used to determine whether an accurate rodent model was completed using the designed model. The airways were visually checked using a video device, and 50 μL of PHMG solution was administered by dividing the PHMG-administered group and the saline-administered group. It was observed that 100% of the lesions occurred in animals that received PHMG (Figures 4A and 5A), with no pathological abnormalities in the saline group (Figures 4B and 5B). The detailed pathological results show that exposure to Polyhexamethylene guanidine (PHMG) can cause pathological changes in lung tissue, including inflammatory cells and interstitial fibrosis. We found the inflammation in the lung tissue, characterized by the infiltration of inflammatory cells such as neutrophils, macrophages, and lymphocytes. These cells can release pro-inflammatory mediators such as cytokines, chemokines, and reactive oxygen species, leading to tissue damage and fibrosis. It causes interstitial fibrosis in the lung tissue, characterized by the accumulation of excess collagen and other extracellular matrix components in the interstitium. Fibrosis can lead to thickening lung tissue and impaired lung function. In addition, it showed the inflammation in the alveoli, the tiny air sacs in the lungs where gas exchange occurs. The infiltration of inflammatory cells such as neutrophils, macrophages, and lymphocytes can lead to alveolar inflammation, characterized by the thickening of the alveolar walls and accumulation of fluid and cells in the alveoli.

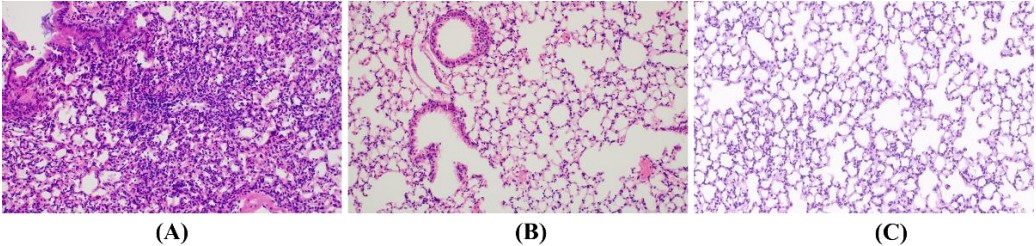

**(A)**            **(B)**            **(C)**

**Figure 4.** Histopathological analysis of lungs in each group. (**A**) A histopathological section of a mouse lung treated with PHMG shows inflammatory cells infiltrating the alveoli (H&E, X200), (**B**) a tissue section of mouse lung treated with PBS, which serves as a negative control (H&E, X200), (**C**) the control group, which did not receive any treatment (H&E, X200) in the mouse model.

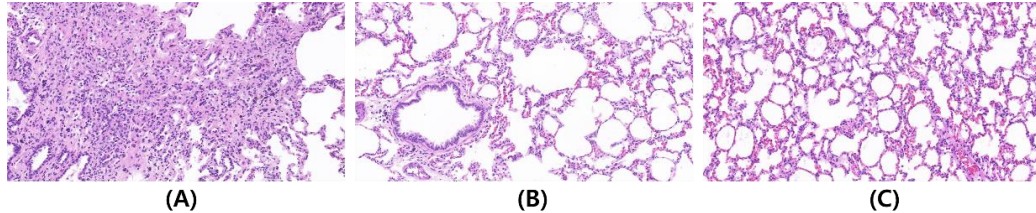

**(A)**            **(B)**            **(C)**

**Figure 5.** Histopathological analysis of lungs in each group. (**A**) A histopathological section of a rat lung treated with PHMG shows inflammatory cells infiltrating fibrosis (H&E, X200), (**B**) a tissue section of rat lung treated with PBS, which serves as a negative control (H&E, X200), (**C**) the control group, which did not receive any treatment (H&E, X200) in the rat model.

## 4. Discussion

The precise administration of drugs and toxic substances is essential to accurately evaluate their effects in animal models. Traditional methods of drug administration, such as oral gavage or intraperitoneal injection, can result in variability in dosing and may not accurately reflect the intended target site. In recent years, a small endoscope has been developed to

precisely administer drugs and toxic substances in small animal models [20–22]. In this discussion, we will evaluate the advantages and limitations of using a small endoscope for accurately administering drugs and toxic substances in small animal models.

Accurate drug administration and precise substance control are vital for the successful design of animal disease models and toxicity evaluation. Various studies have reported intratracheal administration of drugs in rodents. According to Kim et al., a 90% success rate was achieved when constructing an animal model using an automatic video intratracheal drug administration system, where the administration was reported to be possible with an operating time of approximately 1 min [20].

However, a rigid intratracheal drug administration system may provide sufficient images and a suitable route for drug injection in rats. In the case of mice, a smaller and more convenient device is required, considering the narrow intratracheal site and oral structure. Repeated administration in mice should be accompanied by convenient use by the researchers.

There are no reports on the administration of flexible fiber imaging with soluble and powdered substances in any of the previously reported equipment. In this study, a system that can administer powder-type formulations is also one that can be administered to the lungs through an air pump through the powder chamber through the upper respiratory tract.

This study developed a system that can inject a certain amount of liquid and powdered substances with a flexible imaging guide in a microstructure organ than reported systems for intubation. In particular, a system with an outer diameter of less than 2 mm with the fusion of a small imaging probe under 1 mm and a drug injection tip has yet to be reported. Based on these results, the system confirmed 100% administration of toxins to lung lesions in animals such as mice and rats, allowing quantitative administration directly through the airway.

In this study, we evaluated the accuracy of the administration to intratracheal while using only one gender. In the future, we plan to conduct a large-scale survey to identify the response of males and females for toxicity evaluation studies using new toxic substances.

One of the major advantages of using a small endoscope for drug administration is that it allows for precise targeting of the intended site. The use of a small-sized endoscope allows for accurate dosing of the drug or toxic substance. With real-time imaging, researchers can visually confirm the drug's or toxic substance's placement, ensuring precise dosing. Additionally, the use of a small endoscope can reduce variability in drug administration. Traditional methods of drug administration, such as oral gavage or intraperitoneal injection, can result in variability in dosing due to differences in absorption rates or metabolism. With precise administration using a small endoscope, researchers can reduce variability and ensure consistent dosing. Finally, the use of a small endoscope can reduce the number of animals required for the study. With precise administration, researchers can accurately evaluate the drug's or toxic substance's effects in the same animal over time, reducing the need for additional animals.

Research on the lung deposition of liquid or powder using the PennCentury DP-4M inhaler and several self-assembly devices is being actively conducted. Despite the advantages of these systems, several disadvantages have been mentioned by researchers [23]. It has been reported that the powder dispersed in the DP4M inhaler cannot reach deep into the lungs with larger than 5 μm particles, and powder and liquid mass transfer is difficult in small animals, such as mice [24]. There are various devices for delivering drugs to the lungs of rodents, such as the nose and systemic inhalation chambers. However, the primary drawback is the low likelihood of drug absorption. In other words, it is reported that 80–90% of the drug is deposited [25,26]. The drug or toxin delivery to the lungs from the nasal cavity depends on the breathing pattern of the animal. Therefore, a forced blower is required for direct delivery to the lungs in small animal models.

Based on the prevalence of lung disease caused by inhalation exposure due to humidifier disinfectants in the Republic of Korea from 2006 to 2011, this study was conducted

to establish and evaluate the equipment for the evaluation of toxicity in rodent models. The use of an intratracheal drug administration system based on a flexible fiber microendoscope was proposed. PHMG was found to cause lung disease and irreversible fibrosis for one year based on the equipment offered in a previous study [27]. Based on these results, an instrument for evaluating toxic substances such as PHMG is essential. Acute toxicity tests involve exposing organisms to high concentrations of the chemical for a short period, usually 24–48 h. Chronic toxicity tests, on the other hand, expose organisms to lower concentrations of the chemical for a longer period, usually several weeks to several months. Both tests measure the effects of the chemical on the organism's mortality, growth, and reproduction.

Polyhexamethylene guanidine (PHMG) is a biocide commonly used in disinfectants, and as such, its toxicity is a concern for both human health and the environment. Toxicity assessment of PHMG is therefore necessary to ensure its safe use. Toxicity assessment involves a range of tests and experiments to evaluate the potential health and environmental effects of a chemical. These tests may include acute toxicity tests, chronic toxicity tests, genotoxicity tests, and ecotoxicity tests.

Toxicity assessment with PHMG has been carried out in several studies. One study conducted acute and chronic toxicity tests on two aquatic organisms: Daphnia magna and Danio rerio (zebrafish) [28]. The results showed that PHMG was toxic to both organisms, with chronic toxicity being more severe than acute toxicity.

The equipment proposed in this study was a flexible-imaging-fiber-guided intratracheal administration device that can be inserted into micro-organs to administer drugs and toxic substances. In addition, Micro-sized endoscopes are feasible and have great potential for revolutionizing medical imaging and diagnosis. As technology advances, we will likely see further developments in this area, leading to new applications and improvements in medical care.

Researchers who are required to test small animal models will likely find that the designed device will become a universal instrument for inhalable material delivery and monitoring at the laboratory level in the future. However, the device needs to be improved to ensure that the powder is perfectly and evenly distributed in the lungs. In animal experiments, drugs or toxic substances are often administered in powder form for several reasons. Powdered drugs or toxic substances can be accurately weighed and measured, ensuring precise dosing of the test animals. Additionally, some drugs or toxic substances may be less stable in liquid form, which could affect their effectiveness or toxicity. In contrast, powders are typically more stable and less prone to degradation. Finally, administering drugs or toxic substances in powder form can help avoid contamination of the substance, which could affect the results of the experiment. In the future, our research team plans to study using many experimental animals to compare the toxic response to substances in the form of solution and powder substances with the introduced systems.

## 5. Conclusions

This study aimed to design and validate a device such as a flexible-imaging-fiber-guide-based intratracheal drug intubation tube. Several researchers have used animal models to evaluate drug toxicity and efficacy. However, a flexible imaging guide system for small animals is yet to be reported. Overall, the imaging-fiber-guide-based intratracheal drug intubation system may be easily used by researchers and be administered quantitatively, indicating the potential wide use in drug development and toxicity evaluation.

**Author Contributions:** Conceptualization, S.H.J., C.K., J.-H.L. and J.K.; Data curation, H.L. and Y.-W.B.; Formal analysis, S.H.J., C.K., H.L., Y.J.N., C.h.O., Y.-W.B., J.L. and J.-H.L.; Funding acquisition, J.-H.L. and J.K.; Investigation, C.h.O., Y.-W.B. and J.-H.L.; Methodology, S.H.J., C.K., Y.J.N., Y.-W.B., J.L. and J.K.; Project administration, J.-H.L. and J.L.; Resources, S.H.J., H.L., Y.J.N. and J.L.; Supervision, J.K.; Validation, H.L., C.h.O. and J.L.; Visualization, C.K., Y.J.N., C.h.O. and J.-H.L.; Writing—original draft, S.H.J., C.K. and J.K.; Writing—review and editing, J.-H.L. and J.K. All authors have read and agreed to the published version of the manuscript.

**Funding:** This study was supported by the Korea University Grant, Basic Science Research Program through the National Research Foundation of Korea (NRF), funded by the Ministry of Education, Science, and Technology (NRF-2022R1I1A1A01071220), the National Institute of Environment Research (NIER), the Ministry of Environment (MOE) of the Republic of Korea (NIER-2022-04-03-001), the Korea Ministry of Environment as "the Environmental Health Action Program" (2018001360002), and the National Research Foundation of Korea (NRF-2018R1D1A1B07049989).

**Institutional Review Board Statement:** Not applicable.

**Informed Consent Statement:** Not applicable.

**Data Availability Statement:** Not applicable.

**Conflicts of Interest:** The authors declare no conflict of interest.

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
