# Peer review of "Flexible-Imaging-Fiber-Guided Intratracheal Intubation in Rodents"

_applsci, doi:10.3390/app13074253_

Round 1

Reviewer 1 Report

In this manuscript, the authors develop an imaging guide-based drug injectable system, which efficiently delivers PHMG to the lung. While the outcomes are good in general, the conclusion of the study is weakened by several important factors.

Major points:

1.      Page 5 of 10, the material should include the source of PHMG and PBS, and mention whether they use the same PBS to prepare the PHMG. There are two groups in the study (one with PHMG injection and one with PBS injection). Is it better to add one blank control (one mouse and one rat will be good enough) without any liquid injection?

2.      Page 5 of 10, before H&E staining, how do the authors monitor the changes in the lungs? What’s the golden stand to sacrifice the mice?

Minor points:

1.      Page 3 of 10, the animal model is only generated via single gender.

2.      Page 5 of 10, the authors mentioned “per previous studies” without appropriate citations.

3.      Page 5 of 10, the “vocal code position” should be the “vocal cord position”.

Author Response

Once again, we are grateful that the reviewer acknowledged the improvements in the manuscript, and we thank the editor and the reviewer for their additional, very constructive comments and criticisms. We have revised our manuscript as per the recommendations of the reviewers.

Point-by-point responses to your comments and changes made when revising the manuscript are below. The revised manuscript has been submitted to the website.

Here is a point-by-point response to the reviewer's comments and concerns.

We hope this modification meets your approval.

Reviewer 2 Report

This paper represents a reasonable contribution to the field of small animal endoscopy. The novelty here is the use of a flexible endoscope whose main benefit is to improve usability from the point of view of the operator.  Rigid endoscopes have already been demonstrated for rats. The authors suggest that the flexibility will be useful for improved access to the mouse trachea, but the mouse trachea is likely smaller than the rigid probe tip (about 2cm long, according to Figure 1). From the perspective of a mouse, then, the endoscope is not flexible. I give the paper an overall average to low rating, but it does indicate paths for future research (e.g. testing in mice, further miniaturization, evaluation for powder delivery) The authors should discuss the potential for further miniaturization.

The presentation can be improved substantially.

At the end of section 1, the authors imply that rodents are delicate. I am under the impression that rats, anyway, are very tough and resilient. 

Figure 1A, the legend should show 1mm diameter, to eliminate confusion with radius. 1D: the contrast is low (blue on blue) so it is difficult to see the fiber tip. The figure caption has an error: 1B is not a schematic, it is a photograph

Figure 2. What is this device? Is the rat attached to it? Is it for administration of  anesthesia? What is the scale? The caption need to be much more descriptive.

Section 2.3 The PHMG was prepared as per previous studies: a reference is needed here. How are the vocal cords opened?

The first part of the results section more properly belongs in the methods section. The anesthetic concentration quoted here ( 4-5% ) contradicts that mentioned above in section 2.3 (3%). Please reconcile this discrepancy.

Is the inhalation stand depicted in Figure 2? If so the figure should be referenced. How is the rat attached to the stand? 

Typo: vocal cord for "vocal code"

Figure 3. In the figure caption, please describe what we see in the images. Do they show the vocal cords? Are they open?

Discussion paragraph 1. do you mean to write "... not accurately reach..."?
The authors claim that the system can inject powdered substances, but there  are no data offered to support this claim. If there is none, powder administration should be mentioned only as future research. 

Author Response

(The authors gave the same response as above.)

Round 2

Reviewer 1 Report

The authors addressed most of the comments, except for Main Point 1 "Page 5 of 10, the material should include the source of PHMG and PBS, and describe whether they use the same PBS to prepare the PHMG. "

Author Response

Once again, we are grateful that the reviewer acknowledged the improvements in the manuscript, and we thank the editor and the reviewer for their additional, very constructive comments and criticisms. We have revised our manuscript as per the recommendation of the reviewer.

Here is a point-by-point response to the reviewer's comments and concerns.

We hope this modification meets your approval.

We are Sorry for the confusion, the source of the materials information is included in the 2.3 section, and the same PBS was used when preparing the PHMG solution. The contents are as follows.

"An aqueous solution of PHMG (CAS No. 89697-78-9) was prepared by diluting it to 0.9 mg/kg with saline buffer (PBS, Thermo Fisher Scientific Inc., Waltham, MA, USA) as per previous studies [17- 19]."

"For the control group, 50 uL of saline buffer, the same PHMG mixture, was intratracheally administrated to each group with the proposed imaging guide system."
